# 100 Years since the Discovery of Insulin, from Its Discovery to the Insulins of the Future

**DOI:** 10.3390/biomedicines12030533

**Published:** 2024-02-27

**Authors:** Carmen Lambert, Elias Delgado

**Affiliations:** 1Endocrinology, Nutrition, Diabetes and Obesity Group, Health Research Institute of the Principality of Asturias (ISPA), 33011 Oviedo, Asturias, Spain; 2Faculty of Educational Sciences, University of Oviedo, 33005 Oviedo, Asturias, Spain; 3Asturias Central University Hospital, 33011 Oviedo, Asturias, Spain; 4Faculty of Medicine, University of Oviedo, 33006 Oviedo, Asturias, Spain; 5Centre for Biomedical Network Research on Rare Diseases (CIBERER), Institute of Health Carlos III, 28222 Madrid, Spain

**Keywords:** insulin, diabetes, Frederick Banting

## Abstract

The term diabetes first emerged in the 3rd century BC, in a reference by Demetrius of Apamea, who described the disease as a dropsy in which any liquid ingested is eliminated in the form of urine. However, the great discovery that revolutionized this field came from the Canadian doctor Frederick Banting, who together with his student and assistant Charles Best, managed to isolate insulin and treat a patient with diabetes on 23 January 1922. This patient was Leonard Thompson, and the results obtained from him were surprising. His glycosuria and ketonuria disappeared and his blood glucose returned to normal. He received daily injections and lived 13 more years. Advances in the treatment of diabetes have been numerous in the 100 years since its discovery. In this review, we recapitulate the most important events that have occurred, and where research is progressing today.

## 1. History of Diabetes

The first references to diabetes date back to the Egyptians in 1500 BC who described it in “*The Ebers Papyrus*”, a new disease characterized by weight lost, continuous hunger, abundant urination, and an enormous thirst [1]. Many other ancient cultures also described this disease, including the Hindus, ancient Chinese, Greeks, and Romans, among others [2]. In fact, it was the Greek doctor Demetrius of Apamea (3rd century BC), who described the disease as a dropsy in which any fluid that is drunk is discharged as urine, and named it for the first time as Diabetes [3].

Many discoveries were made in the 19th century. In 1815, Michel-Eugene Chevreul identified glucose in urine; a few years later, in 1869, Paul Langerhans described nests of cells in the pancreas; these were later named the “Islets of Langerhans” by Gustave-Edouard Laguesse in 1983 (Figure 1). Twenty years later, in 1889, Von Mering and Minkowsky, two physiologists from the University of Strasbourg, removed a pancreas to test it in a living organism (dog) and observed that its absence induced an excess of sugar in the urine, and subsequently, they found that the pancreas secreted something that reduced the levels of sugar [4,5]. A few years later, in 1891, Eugène Gley confirmed this discovery and revealed that the atrophy of the acinar pancreas did not result in experimental diabetes, and looked for pancreatic extracts which reduced glucosuria; however, Gley did not publish his results [5]. Instead, it was Georg Ludwig Zuelzer who was the first to publish the use of pancreatic extracts, which he called “Acomatol”, on diabetic dogs. He also tried to use Acomatol to treat eight diabetic patients, observing variable reductions in glycosuria and ketonuria, but this also resulted in very severe side effects [6].

During the first half of the 20th century, many physiologists focused their attention on the field of diabetes, one of which was Nicolas Paulesco, who carried out several experiments on pancreatectomized dogs and in 1916 isolated an aqueous pancreatic extract, showing the disappearance of diabetes in pancreatectomized dogs when injected with said extract. Paulesco called this substance “pancreine” and obtained a Romanian patent on the 12 April 1922. However, despite attempts to purify it, pancreine was too toxic to be used in humans [8].

## 2. The Discovery of Insulin

Frederick Banting was born on 14 November 1891 in Alliston, Ontario. He graduated from the school of medicine in 1916 and after completing his specialization in Orthopedics at the Hospital for Sick Children in Toronto, he moved to London, Ontario, and opened a general medicine clinic. Simultaneously, he accepted a part-time position as a professor of surgery and physiology at London’s Western University [2,9]. On November 1st he had to give a course on carbohydrate metabolism. The day before, he was preparing his talk and went to bed after reading the article “The relation of the islets of Langerhans with special reference to cases of pancreatic lithiasis” by pathologist Moses Barron [2,10], a clinical professor of Medicine at the University of Minnesota, published in the latest issue of the journal Surgery, Gynecology and Obstetrics, where he reviewed the pathology of the pancreas and the effects of Wirsung duct ligation on diabetic patients suffering from the obstruction of the pancreatic ducts by stones, and recalled the gradual atrophy of the acini in contrast to the islets of Langerhans. Fascinated by this article, he woke up with an idea at 2.00 a.m. on 31 October 1920 and wrote: “*Diabetus. Ligate pancreatic ducts of dog. Keep dogs alive till acini degenerate leaving islets. Try to isolate the internal secretion of these to relieve glycosurea*” [11,12]. With no proper research training, Banting moved to the University of Toronto for a summer internship, where he worked with Prof. John Macleod who provided Banting with laboratory space, equipment, dogs and a student assistant, Charles Best.

On 17 May 1921, Banting and Best began their experiments with erratic results. Then, Banting began to believe that the islets were formed before the exocrine, and used fetal pancreases as a source for extractions. The first positive results were obtained on 18 November, when the pancreatectomized dog No. 33, called Majorie, was intravenously treated with 10 cc of fetal calf pancreas extract. Her urine became sugar-free within an hour, and she lived for 70 days [8,13]. Their objective was then to produce sufficient stable quantities of pancreatic extract, for which they had the help of the biochemist Dr. James Collip. In order to extract the insulin before it was digested by pancreatic enzymes, they used an extraction method based on the use of varying concentrations of alcohol, which were slightly acidic and kept at a low temperature, and they managed to inactivate the pancreatic enzymes [2,8,13,14]. In this way, Banting and Collip succeeded in producing a pancreatic extract with sufficient potency and purity for human consumption (Figure 2).

After the first failed attempts in humans, on 23 January 1922, the new extract purified by Collip was administered subcutaneously to Leonard Thompson, and the results were spectacular; glycosuria and ketonuria had disappeared, and the blood glucose became normalized. Daily injections of this new extract enabled Leonard to live 13 more years (Figure 3) [15]. However, they did not record the steps they had taken, so it was not until the spring of 1922 that the team was able to successfully produce insulin again. Although they were capable of synthesizing insulin, large-scale production was not feasible for the Toronto group. It was then that Collip and Banting shared their methodology with George H. A. Clowes of Eli Lilly and Company, which had the infrastructure to produce larger quantities of insulin. But it was not until autumn when, using isoelectric precipitation, they were finally able to produce purified insulin on a large scale [16]. Banting, Best and Collip patented the discovery and gave it to the University of Toronto for USD 1, as they wanted everyone who needed it to have access to insulin. As Banting famously said: “Insulin does not belong to me, it belongs to the world”.

On 25 October 1923, the Nobel Prize jury granted the award to Banting and Macleod. Then, Banting, angry after seeing Macleod awarded, decided to share his prize with Best, as did Macleod with Collip soon after. For the first time in the history of the Nobel Prizes, none of the winners attended the award ceremony, which took place on 10 December 1923.

### Insulin Arrives in Europe

In the fall of 1922, August Krogh, a professor at the University of Copenhagen, and his wife, Marie, a physician, came to the United States at the invitation of Yale University to give a series of lectures across the country on their medical research after receiving the Nobel Prize in Physiology in 1920 [17]. During this trip, they heard daily reports of people with diabetes being treated with insulin, and Marie Krogh, who had type 2 diabetes, took a special interest in the treatment. In Boston, Dr. Elliot P. Joslin (the first American diabetes doctor) treated Marie with insulin, and the couple decided to contact Professor Macleod to request permission to manufacture and sell insulin in Scandinavia. August Krogh founded the company Nordisk Insulinlaboratorium with his Danish partner, Dr. Hans Christian Hagedorn, and with funding from the Danish pharmacist August Kongsted [17,18].

In 1925, after a heated argument with Hagendorn, the brothers Harald and Thorvald Pedersen, former Nordisk employees, founded their own company: Novo Terapeutisk Laboratorium. Novo and Nordisk became the world’s leading insulin manufacturers and competed with each other for almost 65 years, until they finally merged in 1989 to form today’s Novo Nordisk s.a. [18].

## 3. The Evolution of the Treatment of Diabetes

### 3.1. The Evolution of Insulin Therapy

Since the discovery of insulin in 1922, there have been numerous advances in the treatment of diabetes, with the common goal of trying to bring the blood glucose and insulin levels of people with diabetes into line with those of healthy people.

Specifically, we know that, in people without diabetes, the maximum peak of insulin secretion occurs 1 h after ingestion and returns to normal levels after another 2 h. However, to achieve a normal insulin profile for 24 h and avoid nocturnal hypoglycemia in diabetic patients, it is necessary to use other insulin formulations that prolong the duration of action.

In this way, we have different types of insulin, with different release times, which allow the patient to better control their disease. Among the different types of insulin, two stand out: rapid-acting insulins, whose action is practically immediate after administration and is short-lived; and long-acting insulins, whose effect takes longer to occur, but lasts longer (Figure 4) [19].

At first, after its discovery, the insulin used was the so-called “amorphous or regular insulin”, but it was an insulin that was not very stable and had many impurities. In 1926, John Jacob Abel, professor of pharmacology at the Johns Hopkins School of Medicine, managed to crystallize amorphous insulin for the first time by adding acetic acid pyridine and zinc, which caused crystals to form, and gave it greater stability, with a slightly later onset of action and producing fewer allergic reactions. This new crystallized insulin quickly displaced amorphous insulin [20]. However, although effective, both insulins required a large number of daily injections. It was then that, in 1936, the first long-acting insulin, the protamine insulin, appeared. Hans Christian Hagedorn, Birger Norman Jensen, Ingrid Wodstrup-Nielsen and Niels B. Krarup published their work, wherein they demonstrated that insulin, together with protamine (a protein from the histone group, obtained from the semen of river trout), increased its action time to almost twenty-four hours. They made the presumption that the greater the insolubility that protamine gave it, the more useful it would be in treatment [21]. The problem was that the solution had to be buffered just before injecting it, making its administration more complicated. Despite this, protamine–insulin boosted the search for new methods to increase the action time of insulin. The next step was the use of metals that would help stabilize the molecule, with nickel and cobalt being the first to appear; however, they did not seem valid. The solution was obtained by Canadians Albert Madden Fisher and David Alymer Scott in 1935, using the most abundant metal in the pancreas, zinc [22]; 10 years later, in 1946, Nordisk was able to form crystals of protamine and insulin using zinc insulin (PZI), and marketed it in 1950 as NPH insulin. NPH insulin had the advantage that it could be mixed with regular insulin directly in the same syringe and had a faster and longer-lasting action than its predecessor [23].

In 1965, Novo (Novo Industri A/S, Bagsværd, Denmark) once again focused its attention on rapid insulins, a field without any progress for around 30 years, and introduced two new insulins: the rapid-acting insulin Actrapid^®^ and the intermediate-acting insulin Crystal II [24]. Researchers already knew that, for many patients, crystalline or regular insulin took too long to act, and a more immediate-acting insulin was needed. Actrapid^®^, also called neutral soluble insulin, was prepared from recrystallized porcine insulin and had a pH of 7.0, at which porcine insulin crystals are soluble. Therefore, it had a faster action than acidic soluble insulin, which had to go from pH 3.0 to a neutral pH before achieving its biological activity [25]. As for Crystal II, insulin came from recrystallized bovine insulin, whose crystals were less soluble and therefore their action was slower [24]. This is when combined insulin treatments appeared, which alternated the use of slow insulins and rapid-acting insulins during the day.

Until that moment, all insulins that had been produced were of animal origin, requiring approximately 50 pigs to cover the annual insulin needs of a patient. This meant that insulin treatment was only available to a few. That changed in 1953, when the British biochemist Frederick Sanger finally determined, after 10 years of work, the amino acid sequence of bovine insulin. Sanger determined that it had two chains, which he called A and B [26]. Chain A had 21 amino acids and chain B, 30 amino acids (Figure 5a,b). They were joined by two disulfide bridges and there was another of these bridges between the amino acids of chain A. Years later, in 1969, it was Dr. Dorothy Crowfoot Hodgkin who would finally describe the three-dimensional structure of the molecule (Figure 5c,d) [27]. Both received the Nobel Prize in Chemistry in 1958 and 1964, respectively, and their discoveries were the beginning of the race to achieve the synthesis of the insulin molecule [12,13].

Porcine insulin differs from human insulin by one amino acid, and from bovine insulin by three. This small difference was enough for some patients to develop an allergy and forced them to abandon treatment. To solve this problem, in 1980 Hoechst, applying a chemical process called transpeptidization, managed to replace the different amino acid in porcine insulin (an alanine) with the amino acid of the human sequence (a threonine). However, despite having achieved its synthesis, insulin was still very expensive.

In 1973, Cohen and Boyer created the first transgenic bacteria that were capable of expressing a foreign gene, and everything seemed to indicate that this technique could be used for the production of proteins or peptides of medical interest [28]. However, it was still necessary to identify the gene encoding insulin in the human genome, which was achieved in 1977 by W. Gilbert and Lydia Villa-Komaroff [29]. However, another problem still had to be solved: insulin is produced from a single chain that is cut in several places until it becomes two chains joined by disulfide bonds, and while bacteria or yeast could synthesize the precursor, they could not process it, so the result would be useless. To solve this problem, the parallel synthesis of the two molecules separately was proposed, which would later be joined by chemical methods. The first to achieve it were Riggs, Itaura and Boyer in 1977. The first clinical trial was carried out on 17 volunteers in July 1980 at Guy’s Hospital in London, and the process was commercialized by Eli Lilly in consortium with Boyer himself and Genetech in 1982 under the trade name Humulin. That new insulin was cheaper to produce, powerful and safe, since it did not show the problems that animal counterparts produced. It began to be distributed in the early 1980s as a treatment for diabetes, being (once again) the first recombinant protein approved as a medicine [30].

Nowadays, practically all diabetics are treated with some type of recombinant insulin, as numerous analogues have been developed, each with different qualities (delayed effect, more powerful, etc.).

The advantage of genetic engineering is that we are not limited to mere copying, but can improve according to the patient’s needs (insulin with immediate effect during a hyperglycemic shock, for example, or persistent over time) [31]. Thus, in 1996, the first rapid insulin analogue, Humalog^®^, was produced; a form of insulin that, by changing the position of two amino acids, managed to increase the speed of the effect. Soon after, in 2000, the first long-acting insulin analogue, insulin Glargine (Lantus^®^), was marketed by Sanofi-Aventis Germany Ltd. (Frankfurt, Germany) throughout the European Union. This new long-acting insulin evolved into a three-times more concentrated formulation of glargine (Glarginia U300), marketed as Toujeo^®^, and was launched following FDA approval in 2015. Additionally, in 2016, the first second-generation long-acting insulin analogue appeared, insulin degludec, marketed by Novo Nordisk as Tresiba^®^; it had a half-life of 25.4 h, reaching its steady state after 3 days [30,31,32,33,34]. New advances have also appeared in rapid insulins, such as the ultra-rapid insulin Faster Aspart (FiAsp^®^), marketed by Novo Nordisk (Bagsværd, Denmark), which was approved in 2017 both in Europe and the USA. This new insulin is formulated with two excipients: vitamin B3 (niacinamide), which increases local blood flow, accelerating the absorption of insulin at that level, and L-arginine, which acts as a stabilizer, achieving a faster onset of action (4.9 min vs. 11.2 min) and a 35% reduction in the time it takes to reach 50% of the maximum concentration [35] (Figure 6).

Additionally, to investigate insulin’s time of action, combinations of insulin treatment with other hypoglicemiant drugs were also researched. In this line, the American Diabetes Association has recommended, since 2019, the administration of a combination of GLP-1RA (glucagon-like peptide 1 receptor agonist) with basal insulin in type 2 diabetes when GLP-1RA is not enough to achieve an optimum glucose control, and injectable therapy is needed [36]. Basal insulin provides a constant level of insulin throughout the day, helping to control fasting blood sugar levels, while GLP-1RAs act to stimulate insulin secretion, inhibit glucagon release, slow gastric emptying and promote satiety, which leads to better blood sugar control, as well as possible weight loss. Due to the different ways in which both medications act, their combination in a fixed-dose injectable formulation offers several advantages. First, it simplifies the treatment regimen for patients by reducing the number of injections needed each day. Second, it addresses multiple aspects of diabetes pathology, focusing on both fasting and postprandial hyperglycemia, as well as providing potential benefits for weight management. Additionally, the combination of these two classes of medications may have synergistic effects, leading to better glycemic control compared to either therapy alone [36,37,38].

### 3.2. The Evolution of Insulin Therapy, More Than Insulin Analogues

After all the advances that took place in the development of new insulins, the efforts began to focus on administration methods as well as the development of equipment capable of measuring blood glucose levels in a rapid, precise and non-invasive way (Figure 7) [39].

#### 3.2.1. Syringes

The administration of insulin in diabetic patients is carried out subcutaneously. Thus, not only have advances in insulin analogues been important during these 100 years, but so too has the progress in methods of insulin administration. Initially, insulin was administered using reusable metal or glass syringes. These hypodermic syringes were sterilized after each use, yet cases of disease transmission have been documented, including hepatitis, malaria, polio and tuberculosis [40].

It was the New Zealand veterinarian and pharmacist Colin Murdoch who, in 1956, developed the first model of disposable syringe, trying to improve on those used to vaccinate animals [41]. He designed a single-use model that would be sold already loaded with the vaccine. When he presented his idea to the New Zealand Department of Health it was considered too futuristic, and the idea did not progress until years later, when it was sold around the world thanks to the Australian company Tasman Vaccine Ltd. Empty disposable plastic syringes did not came onto the market until 1964, pioneered by the leading American medical instrument company Becton Dickinson, which nowadays continues to manufacture about 5 million syringes a day [39].

#### 3.2.2. Insulin Pens

Despite the great advantages, both in terms of safety and comfort for the patient, that the introduction of disposable syringes offered, advances continued towards their disappearance, with the first insulin prefilled pens appearing in 1985—NovoPen from the company Novo Nordisk. Those pens comprised three parts: a single-click-per-unit dosage system, a cartridge for insulin and a disposable needle. That device allowed the patient greater discretion and freedom and better long-term profitability, translating into better adherence to treatment. In 1989, NovoLet was launched, based on the same technology as the previous NovoPen prefilled pens, but in this case, it was disposable and biodegradable [42].

In the year 2007, Next-Generation Insulin pens, also known as “smart pens”, arrived on the market. These devices contain a multidose memory function that records the date, time and dose of previous administrations, which helps better control the insulin doses administered, helping with the better monitoring of the treatment by the doctor. Furthermore, it is important to highlight that the new models on the market allow the administration of insulin in increments of 0.1 units, starting from a minimum dose of half a unit, helping to better adjust the insulin dose needed at all times.

#### 3.2.3. Continuous Subcutaneous Insulin Infusion-CSII

The first commercially used insulin pump was the “Autosyringe” model in 1978, also known as the “Big Blue Brick” [43]. Despite the great enthusiasm of the medical community for this new instrument, they were large devices with very limited reliability. Therefore, it was not until the 1990s when CSII models could finally be introduced into the treatment routine of patients with type 1 diabetes, thanks to the reduction in pump size, increased safety and greater reliability in the administrations. These systems are capable of simulating the physiological secretion of insulin by islet cells more accurately than is performed with multiple daily injections [39,43].

These insulin infusion systems have been evolving over the past few years, allowing the prior programming of doses, as well as adjustments by the patient to special insulin needs such as exercise, illness, hormonal changes, or other circumstances that are out of the ordinary. These changes can be made precisely in increments as small as 0.05–0.25 units per hour. Different clinical trials have demonstrated that the use of CSII improves the glycemic state, and reduces the insulin dosage as well as the glycemic variability of type 1 diabetes patients [40].

Another important advance in the treatment of diabetes was the appearance of continuous glucose monitors, which allow the glucose in the interstitial fluid to be continuously monitored using a sensor inserted into the subcutaneous tissue. Since the beginning of the 2000s, these systems have been capable of integration with an CSII, combining the benefits of both technologies, so that the administration of insulin by the pump is adjusted with the measurements of the continuous glucose monitor, enabling one able to identify trends of increase or decrease in glucose, thus anticipating the necessary dose of insulin [44].

The latest advances in this field occurred in 2017, with the appearance of hybrid closed-loop systems or automated insulin infusion systems, which integrate three components: a real-time continuous glucose monitor, which measures interstitial glucose at regular intervals; a control algorithm, which can be integrated into the insulin pump; an external device or a mobile application; and finally, an insulin pump that allows its continuous release, increasing significantly the time required of patients and reducing HbA1c levels, as well as the number of sever hypo- and hyperglycemia. Since the implementation of the closed-loop systems, their use has increased exponentially, and, currently, 60% of patients with type 1 diabetes in the US use them [45,46,47].

## 4. The Future of Insulin Therapy

Multiple different insulin formulations have been marketed since its discovery in 1922. Currently, according to the FDA [48], we can find three types of approved and marketed insulin: basal insulins, prandial insulins, and insulin mixtures (Table 1). However, research into new formulations and ways of administration of insulin continues, with the near future holing once-weekly basal insulin analogues, glucose-sensitive insulins (also known as smart insulins), and non-injectable insulins.

### 4.1. Once-Weekly Basal Insulin Analogue

One of the main lines of research that is currently being carried out is the search for very long-acting basal insulins, which allow the elimination of daily basal insulin injections, thus requiring only rapid insulin boluses. This is how insulin icodec was developed, developed by Novo Nordisk and whose effectiveness is currently being studied. Icodec has achieved, through structural modifications and binding to C20 fatty diacids, greater molecular stability, less enzymatic degradation and reduced receptor-mediated clearance, allowing less degradation and increasing its lifetime to 196 h [49,50].

### 4.2. Glucose Sensitivity

Also known as smart insulins, the objective of glucose-sensitive insulins is to act in case of hyperglycemia and cease their action after reaching normoglycemia, thus promoting optimal glucose levels in the body. The first studies on these insulins date back to the 1970s; specifically, in 1979, a modified insulin molecule appeared, linked to a lectin, called concavalin A. This molecule would bind to carbohydrates such as glucose and produce a greater release of insulin from this complex; but its final effect was not the desired one [51,52]. Over the years, other studies have been carried out using different types of insulin encapsulations or polymers intended for subcutaneous deposits, among other options. However, none of these products have been translated into therapeutic practice, mainly due to their slow reaction to increases in glucose, meaning that, today, only two molecules have been tested in humans. The main characteristics that a glucose-sensitive insulin must show in order to be transferred to clinical practice are the selective detection of glycemia, no toxicity or side effects, responsiveness to physiological glycemic ranges, and a quick and efficient reaction, reversible in the case of glucose changes [51,53]. Although to date there has been no molecule found that is capable of meeting all these characteristics, there are several pharmaceutical laboratories that are committed to this line of research.

### 4.3. Non-Injectable Insulins

Another line of research that is currently blooming in different pharmaceutical laboratories is the search for non-invasive routes of insulin administration, among which inhaled, intranasal, oral, buccal, and transdermal formulations stand out.

The first non-injectable insulin administration method to hit the market was inhaled insulin, marketed in 2006 by Pfizer under the name Exubera; however, it was quickly withdrawn from the market due to its adverse effects, mainly coughing and alterations of lung function [53,54]. Over the years, new formulations have come to the market, and although none have yet been able to replace injectable insulin, several inhaled insulins are in the clinical trial phase, and seek to establish themselves as a real alternative to injectable insulin in clinical practice.

Possibly the most promising non-invasive route of insulin administration is the oral route, due to its easy administration, which is why there are multiple pharmaceutical laboratories working in this line. Among the main challenges faced by these laboratories are the poor permeability of the epithelial cell due to its high molecular weight, and the enzymatic degradation of oral insulin by gastrointestinal peptidases, so increasing the bioavailability of oral insulin is considered one of the main challenges [55,56].

We can also highlight the long-acting basal insulin analogue 338 (I338), an insulin modified to be less susceptible to proteolytic degradation in the gastrointestinal tract. Its principle is based on a strong and reversible binding to albumin that gives it a long life, with an average plasma concentration of about 70 h [57]. In relation to oral insulins, a capsule-shaped device has been developed, the self-orienting millimeter-scale applicator, SOMA, which after being ingested adheres to the intestinal mucosa to release insulin, and whose objective is to increase the bioavailability of oral insulin [58]. Although preclinical studies are very promising, further research is needed in this field.

## 5. Conclusions

The idea that woke up Frederick Banting in the early morning of 31 October 1920 marked a paradigm shift, not only in the world of diabetes, but in the field of medicine in general, highlighting the importance of collaboration between clinicians, researchers and pharmaceutical companies seeking to improve the lives of people with diabetes. On the other hand, the refusal of the discoverers to conduct business with its discovery allowed significant progress from the first years, as opposed to new developers who have decided to patent the new insulin formulas.

Despite a century having passed since the discovery of insulin, with consequent advances in diabetes research, we are still very far from being able to achieve a real cure for type 1 diabetes that prevents the autoimmune destruction of pancreatic beta cells, although the lives of these patients improve year after year with the new discoveries in this field.

## Figures and Tables

**Figure 1 biomedicines-12-00533-f001:**
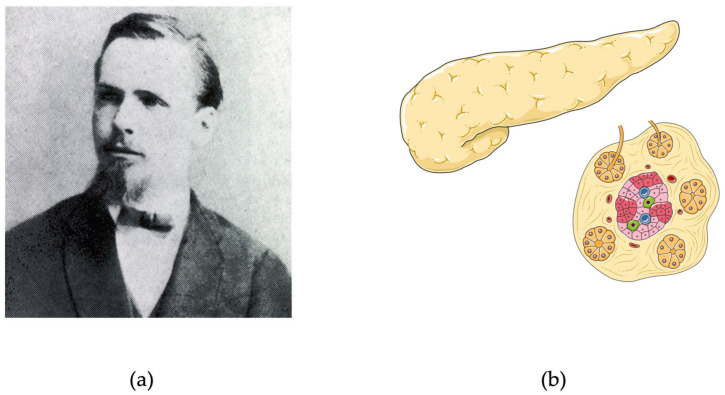
The islets of Langerhans. (**a**) Photograph of Paul Langerhans, discoverer of the islets [7]. (**b**) Drawing of pancreatic islets.

**Figure 2 biomedicines-12-00533-f002:**
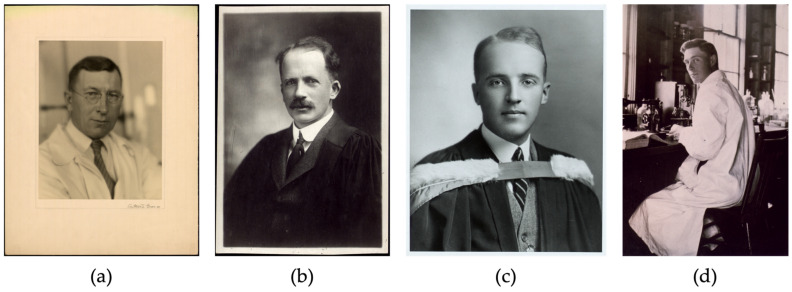
The faces of insulin discovery [7]. (**a**) Frederick Banting, the principal investigator; (**b**) John Macleod, Banting’s supervisor; (**c**) Charles Best, Banting’s assistant and (**d**) James Collip, the individual responsible for developing a method to purify a safe and stable insulin extract.

**Figure 3 biomedicines-12-00533-f003:**
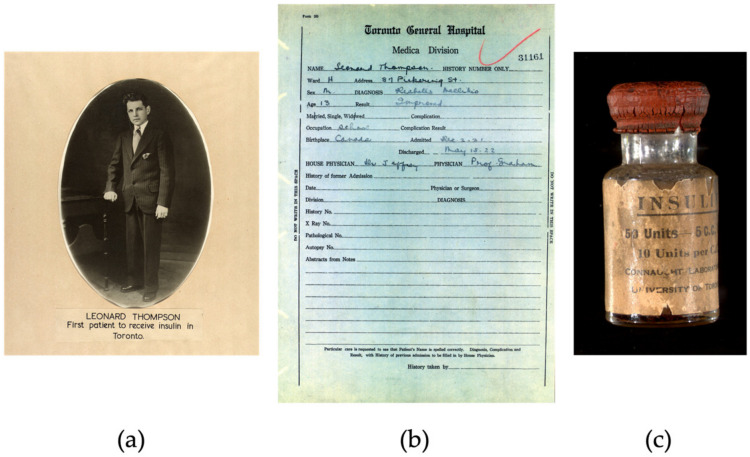
The first successful insulin administration [7]. (**a**) Photography by Leonard Thompson. (**b**) Extract from the patient’s medical record at the Toronto hospital after insulin administration. (**c**) Insulin vial.

**Figure 4 biomedicines-12-00533-f004:**
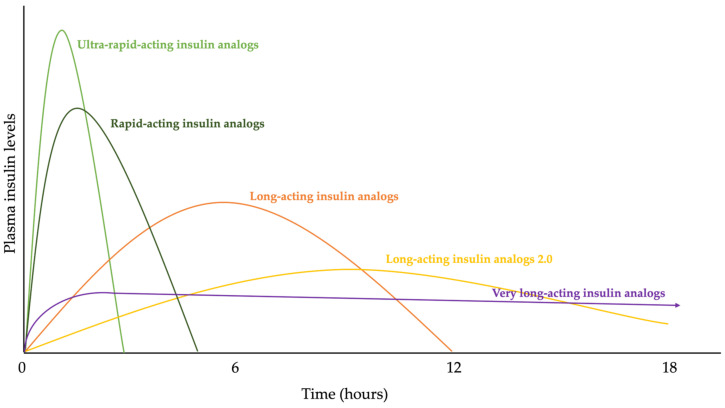
Graph that shows the action time of insulin and its plasma levels according to the type of insulin administered.

**Figure 5 biomedicines-12-00533-f005:**
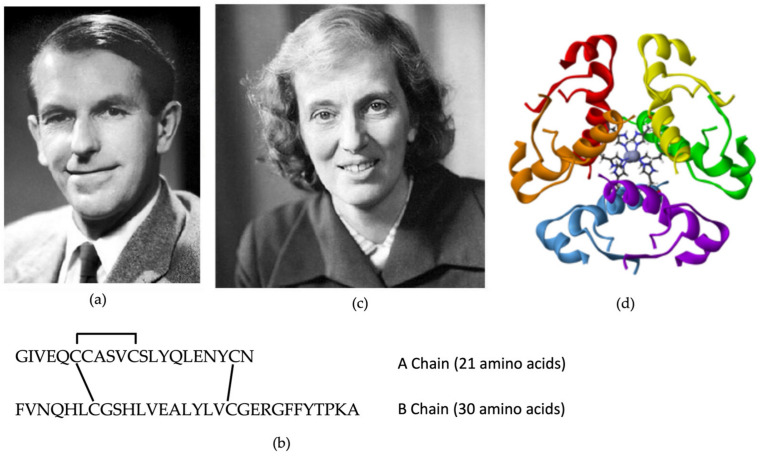
Insulin structure. (**a**) Frederick Sanger, awarded the Nobel Prize in 1958 for the discovery of the amino acid sequence of bovine insulin. (**b**) Amino acid sequence of insulin, made up of two chains, A and B. (**c**) Dr. Dorothy Crowfoot Hodgkin, awarded the Nobel Prize in 1964 for the discovery of the (**d**) three-dimensional structure of insulin.

**Figure 6 biomedicines-12-00533-f006:**
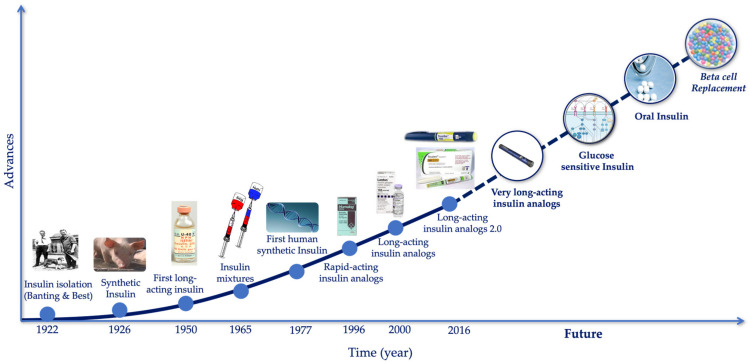
Past and future advances in insulin development.

**Figure 7 biomedicines-12-00533-f007:**
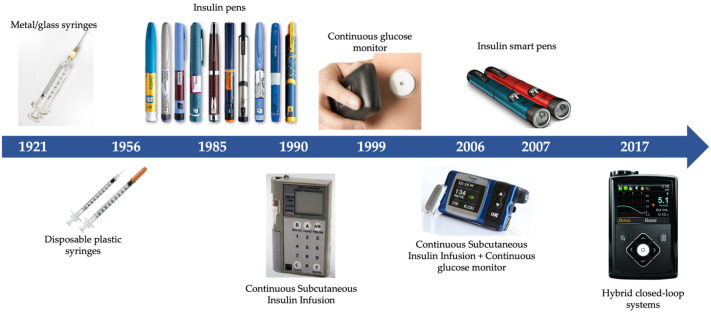
The evolution of insulin administration and glucose monitoring systems.

**Table 1 biomedicines-12-00533-t001:** Different types of insulin approved by FDA.

	Type of Insulin	Brand Name	Other Names
PRANDIAL	Rapid-Acting	Admelog	insulin lispro injection
		Afrezza (inhalation powder)	regular human insulin
		Apidra	insulin glulisine
		Fiasp	insulin aspart
		Humalog	insulin lispro
		Novolog	insulin aspart
BASAL	Intermediate-Acting	Humulin N	NPH human insulin
		Novolin N	
	Short-Acting	Humulin R	regular human insulin
		Novolin R	
	Long-Acting	Basaglar KwikPen	insulin glargine
		Lantus	
		Toujeo	
		Levemir	insulin detemir
		Tresiba FlexTouch	insulin degludec
MIXTURES	Intermediate- and Rapid-Acting	Humalog Mix 75/25	75% insulin lispro protamine suspension
			25% insulin lispro injection
		Humalog 70/30	70% human insulin isophane suspension
			30% human insulin injection
		Humalog Mix 50/50	50% insulin lispro protamine suspension
			50% insulin lispro injection
		NovoLog Mix 70/30	70% insulin aspart protamine suspension
			30% insulin aspart injection
	Long- and Rapid-Acting	Ryzodeg 70/30 FlexTouch	70% insulin degludec
			30% insulin aspart
	Intermediate- and Short-Acting	Humulin 70/30	70% NPH human insulin
			30% regular human insulin injection
		Novolin 70/30	70% NPH Human Insulin
			30% Regular Human Insulin Injection

NPH: neutral protamine Hagedorn.

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
