# Peer review of "100 Years since the Discovery of Insulin, from Its Discovery to the Insulins of the Future"

_biomedicines, 2024, doi:10.3390/biomedicines12030533_

Round 1

Reviewer 1 Report

Comments and Suggestions for Authors

This review highlights the history of the discovery and development of insulin in the past 100 years.

Although this review was written with clarity and summarized the history of insulin in an easy-to-understand way for readers, following issues should be revised by the authors.

1.     Description about ultra-rapid-acting insulin and fixed-dose injection formulation that combine basal insulin and GLP-1RA are needed.

2.     In line 318, the first appearance of TIR should be spelled out as Time in Range.

Author Response

We thank the reviewer for her/his positive comments that improved the manuscript. Below we have addressed (in red) all the comments and incorporated them, also in red, into the new version of our manuscript.

This review highlights the history of the discovery and development of insulin in the past 100 years.

Although this review was written with clarity and summarized the history of insulin in an easy-to understand way for readers, following issues should be revised by the authors.

  1. Description about ultra-rapid-acting insulin and fixed-dose injection formulation that combine basal insulin and GLP-1RA are needed.
    According to the reviewer suggestion we have add a new paragraph in the manuscript describing the ultra-rapid-acting insulin and fixed-dose injection formulation that combine basal insulin and GLP-1RA (lines 248-270).
  2. In line 318, the first appearance of TIR should be spelled out as Time in Range.
    Amended

Reviewer 2 Report

Comments and Suggestions for Authors

This review "100 years since the discovery of insulin. From its discovery to the insulins of the future" is very interesting because it describes the details of the discovery of insulin, as a key therapy for DM.

Comment:

In Paragraph 4. The future of insulin therapy-add Table with key types of insulin therapy.

Author Response

We thank the reviewer for her/his positive comments that improved the manuscript. Below we have addressed (in red) all the comments and incorporated them, also in red, into the new version of our manuscript.

This review "100 years since the discovery of insulin. From its discovery to the insulins of the future" is very interesting because it describes the details of the discovery of insulin, as a key therapy for DM.

Comment:
In Paragraph 4. The future of insulin therapy-add Table with key types of insulin therapy.
According to the reviewer suggestion we have add a new table (Table 1, line 353) summarizing the key types of insulin approved by FDA.

Round 2

Reviewer 1 Report

Comments and Suggestions for Authors

I have no further comment.